# Use of Permanent Wall-Deficient Cells as a System for the Discovery of New-to-Nature Metabolites

**DOI:** 10.3390/microorganisms8121897

**Published:** 2020-11-30

**Authors:** Shraddha Shitut, Güniz Özer Bergman, Alexander Kros, Daniel E. Rozen, Dennis Claessen

**Affiliations:** 1Origins Centre, Nijenborgh 7, 9747 AG Groningen, The Netherlands; 2Institute of Biology, Leiden University, 2333 BE Leiden, The Netherlands; g.ozer.bergman@umail.leidenuniv.nl (G.Ö.B.); d.e.rozen@biology.leidenuniv.nl (D.E.R.); 3Leiden Institute of Chemistry, Leiden University, 2333 CC Leiden, The Netherlands; a.kros@chem.leidenuniv.nl

**Keywords:** secondary metabolites, actinomycetes, protoplast fusion, novel compound discovery, cell wall-deficiency, heteroploidy

## Abstract

Filamentous actinobacteria are widely used as microbial cell factories to produce valuable secondary metabolites, including the vast majority of clinically relevant antimicrobial compounds. Secondary metabolites are typically encoded by large biosynthetic gene clusters, which allow for a modular approach to generating diverse compounds through recombination. Protoplast fusion is a popular method for whole genome recombination that uses fusion of cells that are transiently wall-deficient. This process has been applied for both inter- and intraspecies recombination. An important limiting step in obtaining diverse recombinants from fused protoplasts is regeneration of the cell wall, because this forces the chromosomes from different parental lines to segregate, thereby preventing further recombination. Recently, several labs have gained insight into wall-deficient bacteria that have the ability to proliferate without their cell wall, known as L-forms. Unlike protoplasts, L-forms can stably maintain multiple chromosomes over many division cycles. Fusion of such L-forms would potentially allow cells to express genes from both parental genomes while also extending the time for recombination, both of which can contribute to an increased chemical diversity. Here, we present a perspective on how L-form fusion has the potential to become a platform for novel compound discovery and may thus help to overcome the antibiotic discovery void.

## 1. Introduction

Microorganisms have been used as a chassis to produce beneficial compounds like antibiotics, growth promoters, enzymes, inflammatory drugs, and protease inhibitors. Of these, antibiotics have made a tremendous impact on our lives due to their application in clinical and veterinary settings. Considering the global challenge of increased antimicrobial resistance, there is an urgent need for the discovery of new compounds. Antibiotics are produced via complex pathways encoded by large biosynthetic gene clusters (BGCs). These clusters not only include the necessary genes for biosynthesis of the antibiotic, but also those required for regulation of gene expression, export, and resistance to the antimicrobial compound. Streptomycetes are well-known for their ability to produce a large variety of antibiotics and may carry up to 70 BGCs per genome [1]. Notably, many of these gene clusters are silent or only poorly expressed, which so far has limited the successful exploitation of some of these metabolites as new antimicrobial agents. Genetic engineering of these BGCs for optimal production can be challenging, for instance, due to the presence of cryptic functions and the non-tractability of natural strains that harbor these BGCs. This has been one of the reasons behind the void in novel compound discovery with very few structurally new classes of antibiotics being introduced in the market in the past decades [2]. In contrast, genomic data analysis has unearthed 33,351 putative BGCs in 1154 microbial genomes [3]. The biosynthetic capabilities are thus plentiful, but our capacity to harness them is limited.

Protoplast fusion is a commonly used method to introduce a BGC or entire chromosome into a recipient cell for further genetic manipulation or directed evolution approaches. Its application for genome shuffling has resulted in strains with higher productivity, prominent examples being that of clavulanic acid [4], cephalosporin [5], diverse enzymatic activity [6], and novel compounds (indolizomycin [7]). This method has also led to the activation of silent BGCs [8]. However, protoplast fusion has potential disadvantages, such as the requirement for multiple fusion and regeneration phases, a short time frame during which recombination can occur due to the necessity for protoplasts to regenerate their cell wall, and lastly the instability of recombinants. From this perspective, we take a deeper look into the benefits and hurdles of protoplast fusion and then propose the use of permanent cell-wall-deficient forms as an alternative system for novel compound discovery.

## 2. History of Bacterial Protoplast Fusion

The need for protoplast fusion mainly arose from the difficulty of introducing DNA into cells, especially Gram-positive bacteria. The Gram-positive cell wall typically consists of many interconnected layers of peptidoglycan and teichoic acids [9]. Methods like electroporation and conjugation help alleviate the hurdles of passing through the multilayered cell wall but are restricted in terms of size of DNA that can be transformed. Hence, an alternative was found by stripping away the cell wall altogether. The basic set-up of protoplast fusion involves four consecutive steps: (i) protoplast formation, (ii) protoplast fusion, (iii) recombination between chromosomes within the fused cells, and (iv) reversion of protoplasts to walled cells (Figure 1) [10,11,12]. Protoplasts are obtained by treating walled cells with lysozyme, which degrades the major component of the cell wall, peptidoglycan (PG). The subsequent exposure of protoplasts to crowding agents, such as polyethylene glycol (PEG), causes protoplasts to aggregate. This in turn forces their membranes in close proximity with one another, causing them to fuse. Electrofusion and laser-induced fusion are also commonly used methods to obtain fused cells [12]. Fusion is followed by a brief period of recombination, during which the fusants are maintained in an osmotically protected environment, allowing them to regenerate their cell wall without lysis. In order to select for the desired recombinants, the cells are finally grown in a selective medium.

Because protoplast fusion is non-specific, it has been used for recombination within and between different species. Industrial strains like *Lactobacillus* species have been a common target for genome shuffling via protoplast fusion, resulting in increased yield of lactic acid and improved acid tolerance [13]. Within species fusion may also be combined with random mutagenesis to increase genetic variation, followed by screening for the desired phenotype [14]. For instance, improved degradation of the toxic pesticide pentachlorophenol was achieved in *Sphingobium chrolophenolicum* via protoplast fusion generating cells that had become more resistant [15]. Fusion between phylogenetically distant bacteria has also been performed with *Streptomyces griseus* and *Micromonospora* sp. where recombinants displayed characteristics of both parents. More specifically, recombinants had a colony morphology like *Micromonospora* but with the carbohydrate and amino acid utilization abilities of *S. griseus* [16]. Notably, protoplast fusion is not limited to microbial cells, but has also been successfully used with plants to improve particular traits [10,17].

Taken together, protoplast fusion as a means for genome shuffling has provided several new biological activities and strains with improved traits or growth dynamics [18].

## 3. Disadvantages of Protoplast Fusion

The process of protoplast fusion introduces entire genomes into a single cellular compartment, allowing them to recombine. Protoplasts carrying multiples chromosomes are typically unable to be sustained for long periods of time and are thus unstable. This instability was observed when the mycaminose-producer *Streptomyces fradiae* was fused with a picronolide-producing *Streptomyces* strain [19]. The fusant transiently produced a novel macrolide antibiotic that was not formed by either parent that was hypothesized to have been due to a diploid state of the fusant. Unfortunately, the ability to produce this new compound was rapidly lost after subsequent culturing. Another disadvantage of fusing protoplasts is the limited period during which the chromosomes can fuse with one another to generate recombinants. This is due to the regeneration of a cell wall, which starts immediately after cells have fused. In some fusions, the efficiency of obtaining a recombinant with activity different compared to the parental lines is not quantitatively the same. Protoplasts of the raffinose utilizer *Micromonaspora carbonaceae* and *Streptomyces griseus* were fused by treatment with PEG [16]. Interestingly, only 2 out of 24 fusants revealed different antimicrobial activity compared to their parents. Moreover, protoplast regeneration can be unsuccessful due to inadequate media supplements, rendering the fusants non-viable. Fusants often die without the addition of sucrose, proline, and microelements in specific proportions to the selection media [20]. Lastly, rebuilding the cell wall in fusants producing cell wall-targeting agents may be complicated. This may lead to lower recombinant efficiencies following protoplast fusion [21]. An alternative method would thus be valuable, especially for generating increased phenotypic diversity and stabilizing fusant strains.

## 4. Cell-Wall-Deficient L-Forms as an Attractive Alternative for Protoplasts

While protoplasts are transiently wall-deficient, many bacteria can be forced into a permanent wall-deficient state as so-called L-forms [22,23]. They are variants of bacteria that have an altered cell wall organization, which when made irreversible on account of mutations in the genome result in stable cell wall deficiency. They may be referred to by other names in literature such as L-phase, L-variants, and L-organisms, with the “L” signifying the place of discovery, i.e., the Lister Institute [24]. Since their discovery, L-form research has contributed to the fields of bacterial cell biology [25,26], biotechnology [27], host-associated symbiotic and pathogenic interactions [28,29], and the origin of life [23,30]. Unlike protoplasts, cell-wall-deficient L-forms are able to propagate without their cell wall [31]. Proliferation of L-forms is based on biophysical principles, whereby an imbalance in the cell surface-to-volume ratio of cells causes membrane blebbing, tubulation, and vesicle formation. The imbalance required for this mode of proliferation can be obtained by increasing synthesis of fatty acids [32], which act as building blocks for the membrane, and by increasing the fluidity of the membrane [33]. Many (if not all) bacteria can be forced into an L-form state, including antibiotic-producing actinobacteria [34,35,36,37,38,39,40]. In most cases, loss of the cell wall is triggered by exposing walled cells to a combination of lytic enzymes and antibiotics in an osmotically protective environment. However, formation of wall-less cells has also been observed as a mechanism to escape stress [41], such as osmotic stress in filamentous actinomycetes [38], heat stress in *E. coli* [42], and nutrient stress in *Mycobacterium bovis* [43]. One of the important consequences of wall deficiency is that the resulting cells are polyploid. Because DNA segregation is likely unregulated during this phase, division of L-forms leads to the uneven distribution of chromosomes among progeny (Figure 1). While some cells inherit a single chromosome, others will inherit multiple chromosomes, with different ratios of the parental chromosomal types.

The structural similarity of L-forms to protoplasts makes then amenable to fusion using PEG or electrofusion. Increased membrane fluidity of L-form cells may also facilitate fusion in the absence of such inducing conditions. Indeed, we have observed spontaneous fusion between L-forms (our unpublished results). In contrast to protoplasts, L-form fusion is not followed by regeneration of the cell wall and the subsequent generation of haploid cells (Figure 1). Three potential outcomes are possible after fusion: (i) recombination between chromosomes, (ii) polyploidy where multiple copies of the recombinant genome are present, and (iii) heterokaryosis where both parental genomes are maintained. L-forms could thus overcome the problem of the short recombination time and fusants that only transiently produce a new metabolite due a diploid state prior to cell wall regeneration [19]. The use of L-forms instead allows for maintenance of heterokaryotic genomes in the same cytoplasmic compartment thereby prolonging opportunities for genetic variation to arise through recombination, resulting in novel production phenotypes. The presence of multiple chromosomes can have beneficial effects in terms of productivity (antibiotics, enzymes), fermentation, and biocontrol. This can happen via various routes such as (i) masking of deleterious alleles by a dominant or complimentary copy, (ii) interactions between genes that were previously not in the same environment, and (iii) gene dosage effects (multiple copies of a biosynthetic gene). It has been shown for *Streptomyces* specifically that interactions between different species can induce antibiotic production and activity [44,45]. By introducing genomes of different species in the same compartment using L-forms, one could potentially achieve a similar effect. Metabolite-inducing interactions that are governed by transcriptional regulators or epigenetic modifiers are good candidates to test [46,47]. The introduction of multiple copies of a gene cluster or induction of gene duplication is already in practice for increased antibiotic production [48,49]. For instance, actinorhodin production by *S. coelicolor* was increased 20-fold after tandem amplification of the *act* gene cluster, resulting in nine copies in the genome [48]. In the context of antibiotic production, L-forms are ideal for production of compounds targeting the cell wall since they are naturally resistant. This reduces loss of biomass during production. L-forms also do not show the formation of inclusion bodies, which can be a useful trait for overproducing proteins that are toxic to the cell [27]. Lastly, L-forms can be easily stored at −80 degrees for extended periods of time. It is yet unclear though if subculturing after revival has an effect on the genomes of the hybrid L-forms.

Collectively, the use of CWD bacteria alleviates most of the disadvantages of protoplast fusion. However, L-form fusion itself does have certain lose ends namely, the time and mutations required to achieve wall-deficient propagation, heterogeneity of fusants, and metabolite production levels. Obtaining a stable L-form strain from a wild-type requires multiple rounds of exposure to cell wall-targeting agents. During this exposure, mutations can arise that help alleviate the stress of surviving and growing without a cell wall such [41,50]. Recent studies are shedding light on the metabolic requirements for maintaining a wall-deficient state in a range of bacteria, thus accelerating the process of generating L-forms. The second issue of fusant heterogeneity when using L-forms can make phenotype screening and quantification difficult. Interestingly, the availability of single cell metabolomics, microfluidics, and encapsulation techniques can help solve this by isolating individual fusants immediately after fusion [51,52]. The fusants can then be tracked over time for certain biochemical activity to identify stable phenotypes. Lastly, studies attempting to use L-forms for overproduction of metabolites have resulted in mixed outcomes depending on the species of L-form used. The production of tetracycline by *S. viridifaciens* L-forms was lower than the walled parent [40], while on the other hand *P. mirabilis* L-forms engineered to express a variety of enzymes, antibodies, and polypeptides were able to produce a higher yield than their walled parent [27,53]. A better understanding of L-form growth and its physiology, division, and chromosome dynamics will help to develop this model as an attractive system for identifying novel compounds.

Taken together by providing the extended opportunities for recombination and maintenance of heterokaryons, the wall-deficient lifestyle provides an opportunity for novel compound discovery. The more we understand about growth, division, and DNA segregation in these systems, the better we can apply them for large scale bioprocessing as well.

## Figures and Tables

**Figure 1 microorganisms-08-01897-f001:**
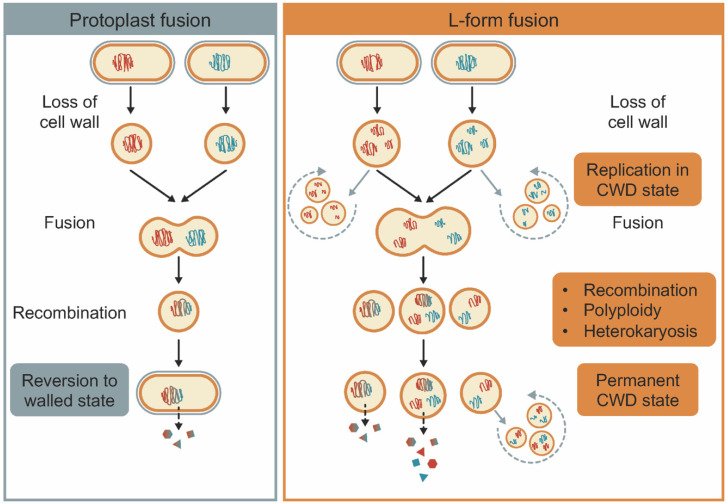
The general process of protoplast fusion compared with L-form fusion. Both methods share some steps like the loss of cell wall and cell–cell fusion but differ in other key steps. Fusion in L-forms can lead to multiple possibilities due to polyploidy in daughter cells, whereas reversion to a walled state is essential in protoplast fusion. Differences in chromosome size are for depiction purposes only and not to scale.

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
