# Peer review of "Use of Permanent Wall-Deficient Cells as a System for the Discovery of New-to-Nature Metabolites"

_microorganisms, 2020, doi:10.3390/microorganisms8121897_

Round 1

Reviewer 1 Report

In this perspective article the authors explore the possibility of using L-form bacteria, namely actinobacteria, for genome recombination as a technology to increase the diversity of BGC and thus identify novel compounds. The ms explores the advantages of using L-form fusion instead of protoplast fusion for genome recombination. This is a very interesting well written ms that has the merit to be published. I just have minor comments:

  • (lines 137-141) The authors present two main routes for which polyploidy could be beneficial in terms of productivity. I believe that gene dosage should also be considered as a feature that could benefit metabolite productivity. It is well known that increasing gene dosage (for instance duplicating a BGC) increases the production of a metabolite.
  • (line 141-143) Induction of antibiotic production through the interactions between different species is often achieved through signalling mechanisms involving small molecules. Please explain how could a similar effect be obtained by introducing genomes of different species in the same L-form bacteria.
  • (lines 144-157) in this paragraph the authors look into the hurdles that L-form fusion still present, particularly concerning BGC evolution and compound discovery. I would also add the fact that metabolite production by L-forms is usually at very low levels. present the L-form bacteri This is a very interesting and well-written paper

Author Response

We thank reviewer 1 for their comments that have helped to improve this manuscript significantly. Please find below a point by point response indicated in blue.

In this perspective article the authors explore the possibility of using L-form bacteria, namely actinobacteria, for genome recombination as a technology to increase the diversity of BGC and thus identify novel compounds. The ms explores the advantages of using L-form fusion instead of protoplast fusion for genome recombination. This is a very interesting well written ms that has the merit to be published. I just have minor comments:

(lines 137-141) The authors present two main routes for which polyploidy could be beneficial in terms of productivity. I believe that gene dosage should also be considered as a feature that could benefit metabolite productivity. It is well known that increasing gene dosage (for instance duplicating a BGC) increases the production of a metabolite.

>>>Indeed, the aspect of gene dosage is important with many examples of gene duplications resulting in increased metabolite production. We have included this point (line 218) as well as examples (lines 223 - 226). We thank the reviewer for pointing this out.

(line 141-143) Induction of antibiotic production through the interactions between different species is often achieved through signalling mechanisms involving small molecules. Please explain how could a similar effect be obtained by introducing genomes of different species in the same L-form bacteria.

>>>Interactions that induce or suppress metabolite production show a range of molecular mechanisms from which signaling molecules are a part. However, interactions are also carried out through transcriptional regulators and other genetic modifiers and are especially involved in awakening cryptic functions which have been found in BGCs. We have included a statement discussing this possibility (lines 221 - 223).

(lines 144-157) in this paragraph the authors look into the hurdles that L-form fusion still present, particularly concerning BGC evolution and compound discovery. I would also add the fact that metabolite production by L-forms is usually at very low levels. present the L-form bacteri This is a very interesting and well-written paper

>>>We thank the reviewer for indicating this potential hurdle. There are unfortunately only a handful of studies investigating the details of metabolite production and secretion from L-forms. We have now included these in the section (line 259 - 263).

Reviewer 2 Report

This is a well written review, although I do have some minor comments.

First, it is a bit short. While conciseness is certainly welcome, this has pared the subject to the bone, and then cut more deeply. Additional context would be useful. There are a number of notable primary papers that appear to be missing, including several early works on L-form bacteria. (Allan, 1991 J Appl Bact; Innes and Allan, 2001, J Appl Bact; Kilcher, et al., 2018, PNAS; etc.) I would suggest the reviews of Gumpert and Hoischen, 1998, Curr Opin Biotech and Grichko and Glick, 1999, Canad J of Chem Eng as places to find some of the early history of this field so that the research can be better placed in context.

To make it easier for general audience to understand the importance of protoplast fusion, I suggest adding a 3-4 sentences around line 59 describing the structure of Gram-positive cell walls and challenges of introducing DNA into bacteria with thick cell walls, like Lactobacillus, Streptomyces, etc.

Line 111: Section on L-forms starts a bit abruptly. Providing some history on how they were developed will be helpful, and may stimulate readers to have additional ideas.

Line 121: “wall-deficiency” should not be hyphenated.

Line 121: “One of the important consequences of wall-deficiency is that the resulting cells are polyploidy”. Is polyploidy merely a consequence of chromosome replication in the absence of cell division? Are there additional mechanisms? What proportion of full genome do the newly recombined genome fragments usually contain? In Fig 1, the L-form fusion diagram, the chromosome of walled bacteria is bigger than the chromosomes in the cell wall deficient state (while in protoplast fusion section chromosome size remains the same). Is this intentional or just for convenience as you prepared the illustration?

Information on long-term storage of L-forms would be helpful. If these alternate forms are going to be used for multiple generations for the production of valuable bioactives, it is necessary to know whether they are amenable to freezing and subsequent recovery, the way that normal bacteria are. Otherwise problems with genetic drift and loss of the synthetic functions seem like an unavoidable consequence.

Author Response

We thank reviewer 2 for the comments and helpful suggestions, it has greatly improved this manuscript. Please find below a point by point response indicated in blue.

This is a well written review, although I do have some minor comments.

First, it is a bit short. While conciseness is certainly welcome, this has pared the subject to the bone, and then cut more deeply. Additional context would be useful. There are a number of notable primary papers that appear to be missing, including several early works on L-form bacteria. (Allan, 1991 J Appl Bact; Innes and Allan, 2001, J Appl Bact; Kilcher, et al., 2018, PNAS; etc.) I would suggest the reviews of Gumpert and Hoischen, 1998, Curr Opin Biotech and Grichko and Glick, 1999, Canad J of Chem Eng as places to find some of the early history of this field so that the research can be better placed in context.

>>>We realize the oversight in terms of referencing however we do intend to keep this article short and concise given its core intent of a perspective. We have included some of the suggested references where relevant (refs 22, 25, 27, 40).

To make it easier for general audience to understand the importance of protoplast fusion, I suggest adding a 3-4 sentences around line 59 describing the structure of Gram-positive cell walls and challenges of introducing DNA into bacteria with thick cell walls, like Lactobacillus, Streptomyces, etc.

>>>We thank the reviewer for this suggestion and discuss this aspect in the revised version (line 103 - 108).

Line 111: Section on L-forms starts a bit abruptly. Providing some history on how they were developed will be helpful, and may stimulate readers to have additional ideas.

>>>This is a good point, however, to keep the focus of this perspective on L-forms as bioproduction chassis we wanted to keep the history of L-forms to a minimum. We do talk briefly about how L-forms have been generated in different species, the various terminology used to describe them and their applications in many fields of biology (lines 181 - 188).

Line 121: “wall-deficiency” should not be hyphenated.

>>>This has been corrected.

Line 121: “One of the important consequences of wall-deficiency is that the resulting cells are polyploidy”. Is polyploidy merely a consequence of chromosome replication in the absence of cell division? Are there additional mechanisms? What proportion of full genome do the newly recombined genome fragments usually contain? In Fig 1, the L-form fusion diagram, the chromosome of walled bacteria is bigger than the chromosomes in the cell wall deficient state (while in protoplast fusion section chromosome size remains the same). Is this intentional or just for convenience as you prepared the illustration?

>>>This is an interesting question, unfortunately there is not much literature on molecular details of chromosomes segregation in L-forms. However, considering the fact that L-forms from a variety of species (both Gram positive and Gram negative) show chromosome replication and cell division we can assume that the absence of a wall is a major factor in the resulting polyploidy. It has also been shown in walled cells using chemical agents that block cell division that chromosomes continue to replicate within leading to polyploidy. In the figure we use a smaller size of chromosomes in the L-forms simply to accommodate the multiple copies, we do not intend it to be to scale and now mention this in the legend (lines 153 – 154).

Information on long-term storage of L-forms would be helpful. If these alternate forms are going to be used for multiple generations for the production of valuable bioactives, it is necessary to know whether they are amenable to freezing and subsequent recovery, the way that normal bacteria are. Otherwise problems with genetic drift and loss of the synthetic functions seem like an unavoidable consequence.

>>>>We thank the reviewer for bringing this aspect to notice and have made amends accordingly (lines 226 - 231).